# Simplified Calculation Model for Typical Dou-Gong Exposed to Vertical Loads

Yiwei Hua 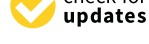, Qing Chun * and Xiaohu Jia

School of Architecture, Southeast University, Nanjing 210096, China; bill_huayw@163.com (Y.H.); archjxh@163.com (X.J.)
* Correspondence: cqnj1979@163.com

**Abstract:** Early Chinese traditional timber buildings preserved until now are mainly ancient buildings built in the time of the Song and Yuan dynasties (960–1368 AD). Dou-gongs of these ancient timber buildings are complex structures. Their complexities, however, are difficult to fully consider in large-scale structural analyses. Therefore, it is necessary to develop a reliable simplified modeling of Dou-gongs, which is applicable for large-scale analyses. In this study, the features of Dou-gongs of early Chinese traditional timber buildings were firstly reviewed, referring to the historical literature and on-site investigation. Then, the mechanical behavior of typical Dou-gongs exposed to vertical loads was examined through refined finite element analyses, where the solid elements were adopted and geometric characteristics were considered. According to the results of the load transferring path, a new beam-truss model representing a simplified Dou-gong was developed, and its accuracy was numerically verified. The results showed that the gravity load of the roof above the column is transferred down through the central axis; the weight of the overhang of the roof is transferred diagonally to the bottom of the Dou-gong, passing through the front of the cantilever components; in the collapse condition, the vertical load is transferred to the two sides through horizontal beams. Compared with the results of the refined model, the new beam-truss model proposed shows an acceptable computational accuracy concerning stress, deformation and stiffness, with 90–97% reduction in the calculation time consumption, which makes it suitable for large-scale structural analyses of early Chinese traditional timber buildings.

**Keywords:** early Chinese traditional timber building; mechanical behavior; Dou-gong; simplified modeling approach; load-transferring path



## 1. Introduction

Dou-gongs play a key role in the traditional timber buildings of most Asian areas, transferring loads from the roof down to the columns. Due to the complex shape of their components and the involvement of various types of ingenious tenon-mortise joints, it is difficult to completely take into account the features of Dou-gongs in large-scale structural analyses. In the recent studies, large-scale structural analyses of the Chinese ancient timber buildings were carried out to investigate their static behavior [1], seismic performance [2–6], and collapse vulnerability [7,8], where Dou-gongs were modeled entirely into a beam, truss, or spring elements. These kinds of simplification usually lack both the consideration of the geometric features of the Dou-gong, as well as of the appropriate numerical calibration or verification. On the other hand, in order to accurately consider the complex features of Dou-gongs, some researchers adopted refined finite element (FE) models in a large-scale analysis [9–12], taking into account both the material properties and geometric characteristics, which are, however, time-consuming in most cases. Thus, it is necessary to propose a reliable simplified approach for the Dou-gong modeling, by considering both construction regulations (such as dimension and connection rules of the components referring to Yin-zao-fa-shi [13]) and high efficiency.

This research aims to propose a new simplified approach to Dou-gong modeling. In this study, the Dou-gongs of Chinese traditional timber buildings built in the time of the Song and Yuan dynasties (960–1368 AD) are chosen, because the Dou-gongs in this period involve various types of the components, with a more complex arrangement and construction (Figure 1a). A simplified model for these complex Dou-gongs can be conveniently extended to other simpler ones. In addition, compared with Dou-gongs built in the Ming and Qing dynasties (1368–1912 AD), as shown in Figure 1b, those built in the time of the Song and Yuan dynasties (960–1368 AD) exhibit a larger dimension, which plays a more critical role in the entire building structure. An unsuitable simplification of the Dou-gongs may produce inaccurate numerical results of the large-scale structural analysis. The style of the Dou-gong investigated in recent works is closer to the one from Japan and northern China, built around the period of the Ming and Qing dynasties (1368–1912 AD), which is quite different from the ones built in the time of the Song and Yuan dynasties (see [9,14–20] as an example). Thus, a refined finite element analysis of the typical Dou-gongs built during the time of the Song and Yuan dynasties will be first carried out for a comprehensive understanding of their stress mechanism and load transferring path.

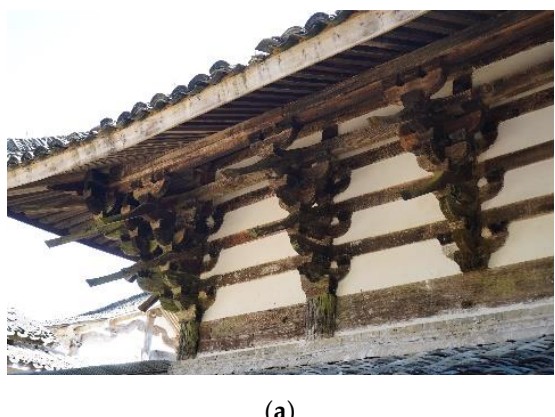 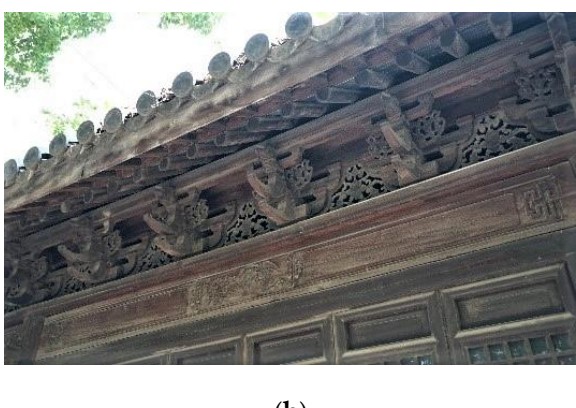

(**a**)          (**b**)

**Figure 1.** Two samples of the Dou-gong from the Yuan dynasty and Ming dynasty: (**a**) Shi-si Temple, Li-shui, Zhe-jiang, Yuan dynasty; (**b**) Wen Miao, Su-zhou, Jiang-su, Ming dynasty.

The frame of this paper is organized as follows: first, the features of Dou-gongs of early Chinese traditional timber buildings were analyzed and summarized through the historical literature review and on-site investigation. Then, typical Dou-gongs were chosen and their refined FE analyses were performed. Their load transferring path and load-bearing mechanism under vertical loads were studied. Based on the obtained results, a new beam-truss model was developed for the Dou-gong simplification, and the accuracy of the model was numerically verified. The proposed simplification procedure can also produce a reference for establishing a simplified model of the Dou-gongs of the ancient timber buildings in other Asian areas.

## 2. Mechanical Behavior of Typical Dou-Gongs

To provide a basis for the simplified model, the mechanical behavior of typical Dou-gongs of early Chinese traditional timber buildings exposed to vertical loads was carefully investigated. In this section, Dou-gongs of early Chinese traditional timber buildings were investigated and their characteristics were summarized. Then, extensive numerical investigations of typical Dou-gongs under two vertical load working conditions were performed, and the transferring path of tensile and compressive stresses was studied based on the refined FE models, where the geometric features of the Dou-gongs and the contact among their components were fully taken into account.

### 2.1. Typical Dou-Gongs of Early Chinese Traditional Timber Buildings

In this subsection, Dou-gongs of early Chinese traditional timber buildings were investigated, based on the literature review and on-site investigation. The arrangement methods of their components and connections with building beams or columns were studied. The obtained results provided a basis for the numerical study presented later.

Some terms describing the parts of Dou-gongs, which were referenced in the literature [21], were adopted herein. As shown in Figure 2a, each tier of a Gong is called "Tiao". From the central axis to the two sides, each tier is termed 1st Tiao, 2nd Tiao, etc., in sequence. The dimensions of the Dou-gong components are termed Cai, Fen, and Zhi. As shown in Figure 2b, Cai, the standard construction timber, is divided into eight classes. Fen, the standard unit for the dimensions of components, changes with different Cai classes (e.g., for the sixth grade Cai in the Song dynasty, 1 Fen is approximately equal to 12.8 mm). The section of the Dou-gong components may be divided into two different types (Figure 2b). A component is termed Dan Cai if its height is equal to a standard Cai's depth (15 Fens); a component is termed Zu Cai if its height reaches the Cai height plus Zhi (6 Fens). Both Dan Cai and Zu Cai share the same width (10 Fens). Some Tiaos adopting Gongs are called Ji-xin, and the arrangement of these Gongs is generally divided into Dan- and Chong-gongs. The former corresponds to the situation when a single Gong is adopted to support the beams above (Figure 2c), whereas in the case of the latter, two Gongs are stacked to support the beams above (Figure 2d). A Tiao without any Gong is called Tou-xin (e.g., 1st Tiao in the rear, Figure 2a). A Dou-gong is called "Ji-xin-zao" if its every Tiao is Ji-xin (Figure 2e), whereas if its every Tiao is Tou-xin, the Dou-gong is named "Tou-xin-zao" (Figure 2f).

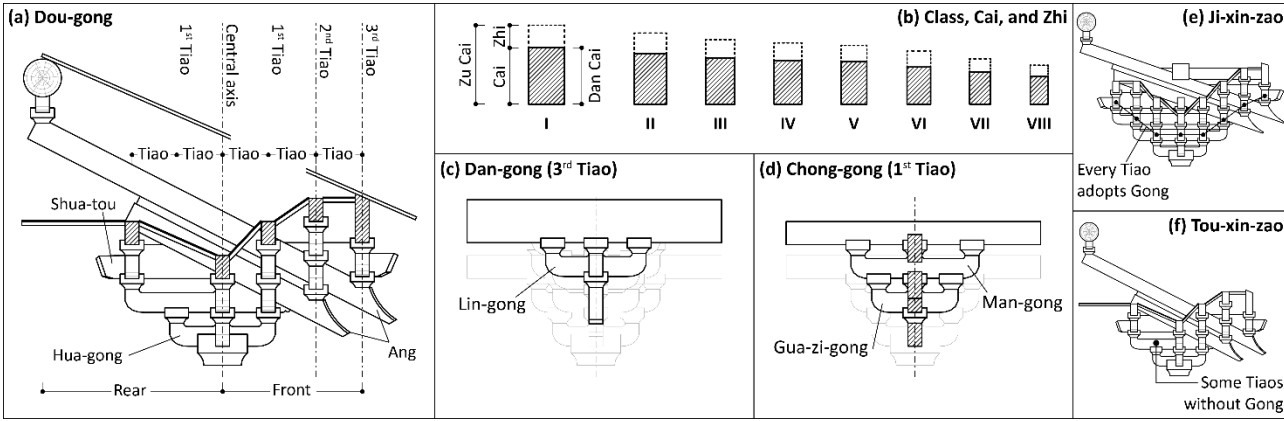

**Figure 2.** Description of Dou-gongs used in ancient China: (**a**) a typical section of Dou-gong, name of terms and components; (**b**) eight classes of section, definition of Dan and Zu Cai; (**c**) Dan-gong: single Gong supports the beam above; (**d**) Chong-gong: two Gongs are stacked to support the beam above; (**e**) Ji-xin-zao: every Tiao of the Dou-gong adopts Gongs; (**f**) Tou-xin-zao: some Tiaos of the Dou-gong do not use Gongs.

Overall, the Dou-gong characteristics of early Chinese traditional timber buildings, six Chinese traditional timber buildings built during the Song dynasty, and four Chinese traditional timber buildings built in the time of the Yuan dynasty, were carefully studied. Among these buildings, Dou-gongs of the main hall of Bao-guo Temple, Yan-fu Temple, Tian-ning Temple, and Shi-si Temple were carefully examined through on-site investigations (Figure 3), and the Dou-gongs of other buildings were investigated by referring to the relevant literature [22–31]. As a result, the features of Dou-gongs adopted in these buildings are listed in Table 1.

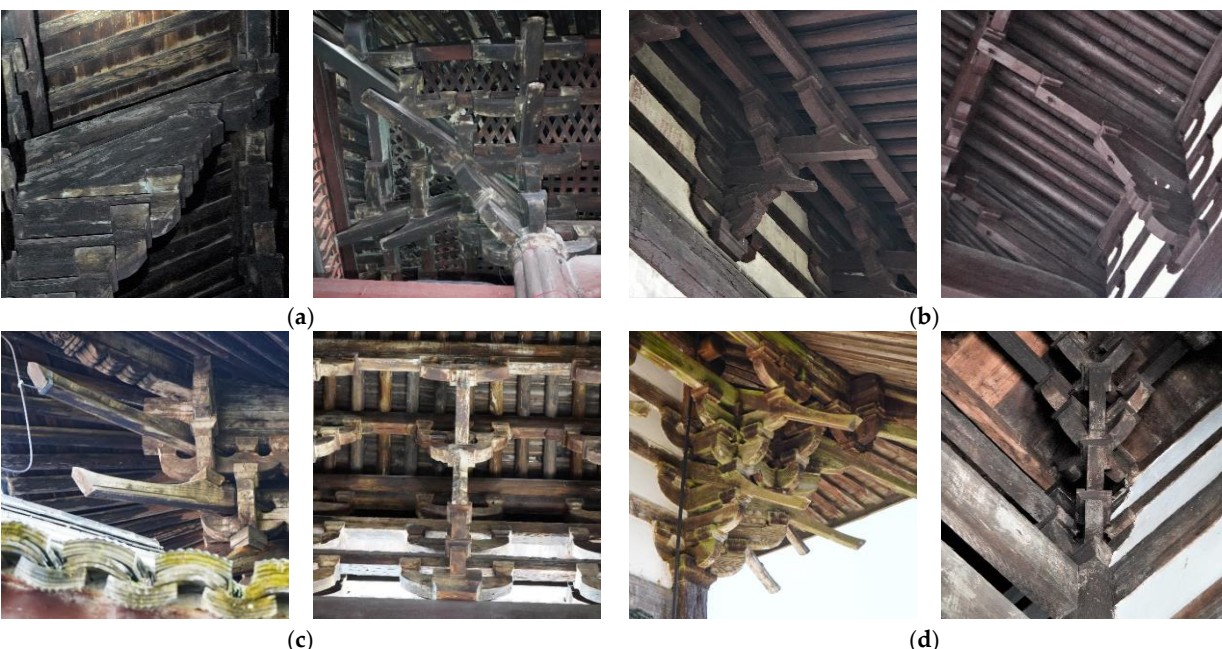

**Figure 3.** On-site investigations of the typical Dou-gongs of early Chinese traditional timber buildings: (**a**) Dou-gongs in the main hall of Bao-guo Temple (AD 1013); (**b**) Dou-gongs in the main hall of Tianning Temple (AD 1318); (**c**) Dou-gongs in the main hall of Yan-fu Temple (AD 1317); (**d**) Dou-gongs in the main hall of Shi-si Temple (AD 1356).

For the six cases in the Song dynasty, five have seven-layer-stacked Dou-gongs. The Dou-gongs in the main hall of the Hua-lin Temple, the main hall of the Bao-guo Temple, and the main hall of the Yuan-miao Temple are quite similar, and are seven-layer stacked with two Hua-gongs and two Angs. The rears of these three Dou-gongs are fully connected with beams, columns, or purlins. Among them, Dou-gongs in the main halls of the Hua-lin Temple and Yuan-miao Temple share more common features, which are as follows: Angs and Hua-gongs have Dan Cais; the 2nd and 4th Tiaos in the front are Jin-xin, using Chong- and Dan-gongs, respectively, and the others are Tou-xin. Compared with the two Dou-gongs above, Dou-gongs in the main hall of the Bao-guo Temple have some differences. Angs and Hua-gongs of the intermediate sets are Dan Cai, while those of the column sets are Zu Cai; the 2nd–4th Tiaos in the front have Dan-gongs. Dou-gongs of the main halls of the Shi-si Temple and Xuan-miao Temple have Zu Cai, while the rears of these Dou-gongs lack effective connections with beams, columns, or purlins behind. Besides five seven-layer-stacked Dou-gongs, there is also a five-layer stacked Dou-gong in the Song dynasty, adopting one Hua-gong and one Ang. The dimensions of the Ang and Hua-gong of the intermediate and column sets are also different, Zu Cai for the column sets and Dan Cai for the intermediate sets. In this sample, most Tiaos are Tou-xin, and the rear part is connected with beams and purlins.

A smaller number of the stacks is adopted in Dou-gongs of the Yuan dynasty, and all these cases have Zu Cais. Two six-layer stacked Dou-gongs show many similarities; both of them adopt two Angs and one Hua-gong, and only the 2nd–3rd front Tiaos are Ji-xin. Compared with the two Dou-gongs mentioned above, the dimension of the Dou-gong in the main halls of the Xuan-yuan Palace (five-layer stacked) and Yun-yan Temple (four-layer stacked) are smaller, where the front Tiaos are all Ji-xin, and the rear Tiaos are all Tou-xin. The Dou-gong of the main hall of Xuan-yuan Palace also has two Angs whose tail is connected with purlins, while Dou-gongs of the gate building of the Yun-yan Temple do not have Angs. All the column sets in the four cases of the Yuan dynasty are connected with beams and purlins, but are not connected with columns.

**Table 1.** Characteristics of Dou-gongs in the ancient Chinese timber buildings from the time of the Song and Yuan dynasties.

| Dynasty | Name of the Building | Location | Style | Cai | Arrangement of the Gong | | | Connections | Diagram | |
|---|---|---|---|---|---|---|---|---|---|---|
| | | | | | Front | Rear | Central Axis | | Col. | Inter. |
| Song | The main hall of Hua-lin Temple [22] | Col. | 7-layer stacked, 2 Hua-gongs, 2 Angs | Dan | 2nd Tiao is Chong-gong (Ji-xin), 4th Tiao is Dan-gong (Ji-xin) | 2nd Tiao is Chong-gong (Ji-xin) | 3 Dan-gongs | Connected with beams, columns, and purlins |  |  |
| | | Inter. | | Dan | | 1st–5th Tiaos are Tou-xin | | Connected with purlins | | |
| Song | The main hall of Bao-guo Temple [23] | Col. | 7-layer stacked, 2 Hua-gongs, 2 Angs | Zu | 2nd–4th Tiaos are Dan-gong (Ji-xin) | 1st–2nd Tiaos are Tou-xin | 2 Dan-gongs and 1 Chong-gong | Connected with beams, columns, and purlins |  |  |
| | | Inter. | | Dan | | 1st–4th Tiaos are Tou-xin | | Connected with purlins | | |
| Song | The main hall of Yuan-miao Temple [24] | Col. | 7-layer stacked, 2 Hua-gongs, 2 Angs | Dan | 2nd Tiao is Chong-gong (Ji-xin), 4th Tiao is Dan-gong (Ji-xin) | 1st–2nd Tiaos are Tou-xin | 2 Dan-gongs and 1 Chong-gong | Connected with beams, columns, and purlins |  |  |
| | | Inter. | | Dan | | 1st–4th Tiaos are Tou-xin | | Connected with purlins | | |
| Song | The main hall of Bao-sheng Temple [25] | Col. | 5-layer stacked, 1 Hua-gong, 1 Ang | Zu | 2nd Tiao is Dan-gong (Ji-xin) | 1st Tiaos is Tou-xin | 1 Dan-gong and 1 Chong-gong | Connected with beams and purlins |  |  |
| | | Inter. | | Dan | | 1st–2nd Tiaos are Tou-xin | | Connected with purlins | | |
| Song | The main hall of Shi-si Temple [26] | Col. | 7-layer stacked, 3 Hua-gongs, 2 Ang | Dan | 2nd Tiao is Chong-gong (Ji-xin), 4th Tiao is Dan-gong (Ji-xin) | 2nd Tiao is Chong-gong (Ji-xin), 4th Tiao is Dan-gong (Ji-xin) | 3 Dan-gongs | Connected with beams |  |  |
| | | Inter. | | Dan | | | | Without any connection with the purlins or beams | | |
| Song | The main hall of Xuan-miao Temple [27] | Col. | 7-layer stacked, 2 Hua-gongs, 2 Angs | Zu | 2nd–4th Tiaos are Dan-gong (Ji-xin) | 2nd–4th Tiaos are Dan-gong (Ji-xin) | 1 Chong-gong | Without any connection with the purlins or beams |  |  |
| Yuan | The main hall of Yan-fu Temple [28] | Col. | 6-layer stacked, 1 Hua-gongs, 2 Angs | Zu | 2nd–3rd Tiaos are Dan-gong (Ji-xin) | 1st–2nd Tiaos are Tou-xin | 3 Dan-gongs | Connected with beams and purlins |  |  |
| | | Inter. | | Zu | | 1st–2nd Tiaos are Tou-xin | | Connected with purlins | | |
| Yuan | The main hall of Tian-ning Temple | Col. | 6-layer stacked, 1 Hua-gongs, 2 Angs | Zu | 2nd Tiao is Chong-gong (Ji-xin), 3rd Tiao is Dan-gong (Ji-xin) | 1st Tiaos is Tou-xin | 3 Dan-gongs | Connected with beams and purlins |  |  |
| | | Inter. | | Zu | | 1st Tiaos is Tou-xin | | Connected with purlins | | |

**Table 1.** *Cont.*

| Dynasty | Name of the Building | Location | Style | Cai | Arrangement of the Gong | | | Connections | Diagram | |
|---------|---------------------|----------|-------|-----|-------|------|--------------|-------------|------|-------|
| | | | | | Front | Rear | Central Axis | | Col. | Inter. |
| Yuan | The main hall of Yun-yan Temple [29] | Col. | 4-layer stacked, 1 Hua-gongs | Zu | 1st Tiao is Dan-gong (Ji-xin) | 1st Tiaos is Tou-xin | 1 Chong-gong | Connected with beams |  |  |
| | | Inter. | | Zu | | 1st–2nd Tiaos are Tou-xin | | Connected with purlins | | |
| Yuan | The main hall of Xuan-yuan Palace [30] | Inter. | 5-layer stacked, 2 Angs | Zu | 1st Tiao is Chong-gong (Ji-xin), 2nd Tiao is Dan-gong (Ji-xin) | 1st–2nd Tiaos are Tou-xin | 1 Chong-gong | Connected with purlins | – |  |

**Note:** In the column of "Location", abbreviation of Col. and Inter. represents "column set" and "intermediate set", respectively. In the column of "arrangement of the Gongs", all the Tiaos not mentioned in the table are Tou-xin. For the Dou-gongs in the main hall of Xuan-yuan Palace, only the information of the intermediate sets was given because only these sets were proved to be built in the Song dynasty [31]. For the Dou-gongs in the main hall of the Xuan-miao Temple, the style of the column set and the intermediate set is the same, whose constructions above the ceiling were not mentioned in the literature [27]. For the Dou-gongs of the main hall of the Bao-guo Temple, Yan-fu Temple, and Shi-si Temple, on-site investigations were conducted, in addition to referring to the relative literature. For the Dou-gongs of the main hall of the Tian-ning temple, their constructions and dimensions were obtained through field measurement and 3D scanning, which will be provided below.

From the summary and analysis above, the changes in the style of Dou-gongs from the Song to Yuan dynasties are obvious. The Dou-gong dimension becomes smaller, and the Cai adopted in the Dou-gong changes from Dan Cai to Zu Cai. Most Dou-gongs in the Song dynasty are seven-layer-stacked with two Hua-gongs and two Angs. Some front Tiaos of Dou-gongs are Ji-xin, and most of the rear Tiaos are Tou-xin. The Dan-gong was commonly used at the central axis. The rear of the Dou-gong is fully connected with beams, columns, or purlins behind. In the Yuan dynasty, the number of the Dou-gong stacked layers is smaller. The Ang remains in these Dou-gongs, while its function is gradually weakened. In addition, in the Yuan dynasty, the connections between Dou-gongs and beams, columns, or purlins are weaker than those in the Song dynasty. Generally, the features of Dou-gongs of early Chinese traditional timber buildings can be listed as follows:

- Most of the Dou-gongs are Tou-xin-zao;
- Angs are commonly used (especially double ones), whose tail reaches the purlins behind;
- The rear part is usually connected with beams or columns.

Dou-gongs of the main hall of the Tian-ning Temple, built in 1318 AD, can be regarded as a typical Dou-gong of the Song and Yuan dynasties. These Dou-gongs have two Angs, whose tails reach the purlins behind; the number of the stacked layers of these Dou-gongs is six, which is between that of the Dou-gongs in the early Song (e.g., the main hall of the Hua-lin Temple) and the later Yuan (e.g., the main hall of the Yun-yan Temple) dynasties.

The Dou-gongs involved in recent studies were generally typical ones from Japan and northern China, built around the time of the Ming and Qing dynasties (1368–1912 AD), which is quite different from the Dou-gongs discussed above (see [9,14,20] as an example). Therefore, the refined finite element analysis of the typical Dou-gong built in the Song and Yuan dynasties is necessary to understand their load-bearing mechanism. As discussed above, Dou-gongs in the main hall of the Tian-ning Temple are selected as a representative of Tou-xin-zao Dou-gongs, and a Ji-xin-zao Dou-gong was designed based on the relevant literature [13,32] for a comparative study. The dimensions of these two Dou-gongs were consistent, and the name of their arranged components is presented in Figure 4. These two Dou-gongs were investigated and compared through the refined finite element analyses in the next subsection.

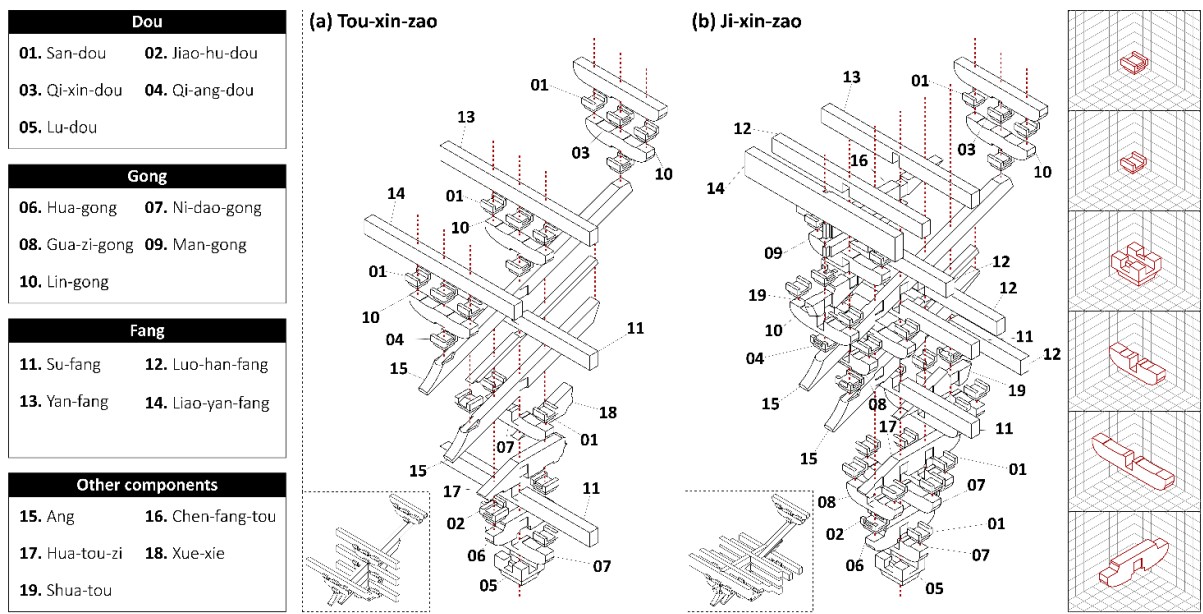

**Figure 4.** Overview of the two Dou-gongs involved in this study.

### 2.2. Typical Dou-Gongs of Early Chinese Traditional Timber Buildings

Refined FE models of the considered Dou-gongs were established based on the on-site 3D scanning and investigation (Figure 5), where most geometric characteristics (e.g., groove, tenon and mortise) were considered. The "hard-contact" model was used in the analysis to simulate normal contacts among the components, which ensures a negligible penetration of the mesh at the contact interface. The tangential behavior of the contact was frictional with a coefficient of 0.3. Referring to the literature [23], the main wood material of early Chinese traditional timber buildings was Chinese fir. So, in this study, the wood material of Dou-gongs was considered accordingly. The orthotropic model was used to simulate the wood mechanical behavior; the elastic modulus parallel to the grain was assumed to be 9000 MPa according to the Chinese code [33], and the elastic modulus perpendicular to the grain was taken as 900 MPa (1/10 of the elastic modulus parallel to the grain, based on the literature [34]). The Poisson's ratio of the wood of 0.3 was used. Considering that the components of the Dou-gong remain elastic in most cases for the vertical loading, as a simplification, the plastic behavior and failure of the wood were ignored in this study. The complex contact among the components generally tends to result in convergence difficulties. To avoid this, quasi-static analyses were conducted through the ABAQUS/Explicit. The "General Contact" technology in ABAQUS was adopted to ensure the behavior of two contacted surfaces among the component to obey the contact law mentioned above. The average time step of the explicit algorithm was around $10^{-6}$ s.

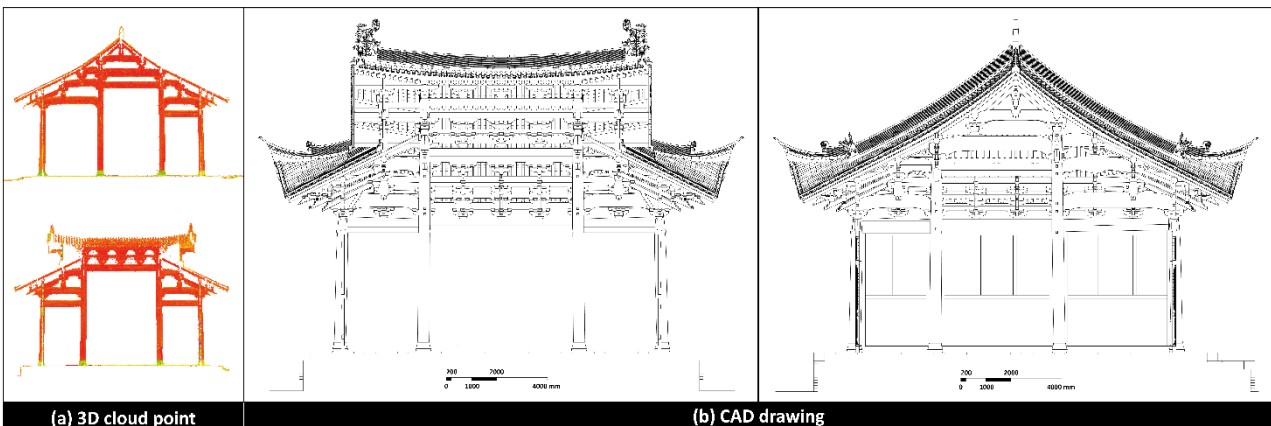

**Figure 5.** Three-dimensional cloud points and the CAD drawing of the main hall of the Tian-ning Temple.

In this study, two different vertical load conditions were considered. One was the roof load condition shown in Figure 6a, which represents the most common situation. To simulate this load condition, as shown in Figure 6b,c, the nodes at the bottom of the Lu-dou were pinned, the nodes at the top of the Lin-gong on the tail of the Ang were vertically constrained, and the nodes at the two sides of Fangs were horizontally constrained. The loading applied to the Liao-yan- and Yan-fangs, aimed to simulate the load on the roof (e.g., gravity of the tiles, mortar and sheathing board), amounted to 2.16 and 9.46 kN/m, respectively and was calculated through a combination of dead (2.4 kN/m$^2$) and live (0.7 kN/m$^2$) loads on the roof, based on the code [35].

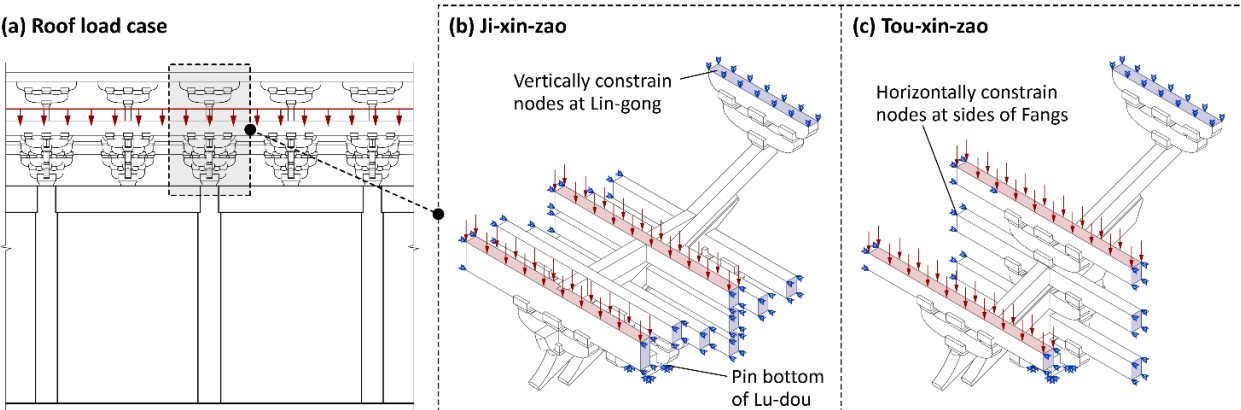

**Figure 6.** Load and boundary definitions of the two finite element models corresponding to the roof load condition. (**a**) Roof load case.

On the other hand, recently, many ancient timber buildings collapse every year in China, due to local failures. For a better understanding of Dou-gong structural behavior, the collapse condition was also involved in this study. In recent literature [36,37], the single-column-removal (SCR) scenario was commonly used to study the structural performance in the progressive collapse, which was also adopted in this study (Figure 7a). This scenario assumes a sudden failure of the middle column, and as a result, an extra load caused by this collapse is vertically added on the top of the structure over the failed column. Therefore, the boundary conditions of two Dou-gongs were set as indicated in Figure 7b,c; the two sides of the Fangs were vertically constrained, and a vertical displacement was imposed simultaneously in the middle of the Yan-fang and at the bottom of the Lu-dou, simulating the extra load caused by the failure of the column below.

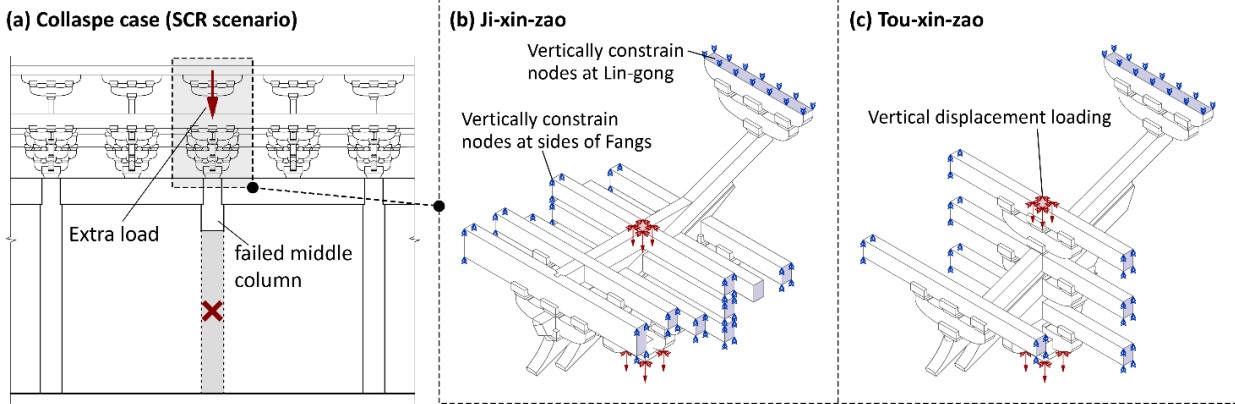

**Figure 7.** Load and boundary definitions of the two finite element models in the collapse condition.

As a result, the diagrams of the maximum principal stress vector of the two refined FE models under the considered load conditions are shown below.

For the Ji-xin-zao Dou-gong bearing the roof load (Figure 8a), the fronts of two Angs bend together, bearing the roof load from the Liao-yan-fang. The middle part is compressed, transferring the roof load from the Yan-fang down to the Hua-tou-zi. The stress at the tails of the Angs is comparatively small, transferring the roof load up to the purlin (support). The stress at the upper side of the Chen-fang-tou and Shua-tou is tensile, while the stress at their bottom surfaces is compressive, indicating that the Chen-fang-tou and Shua-tou are also in bending. The Hua-gong and Hua-tou-zi are generally compressed, where the stress at the front is significantly larger, and the direction of the compressions is perpendicular to the Ang slopes. This indicates that the fronts of the Hua-gong and Hua-tou-zi directly

transfer the compression from the front of the Angs to the Lu-dou. The middle of the Hua-gong and Hua-tou-zi, where the direction of the stress is vertically downward, delivers the load from the middle of the Angs down to the Lu-dou. The Dous in the Ji-xin-zao Dou-gong are locally compressed. Among them, the stress of the Lu-dou is the largest, where the compression is concentrated at the front. The compressive stress of the two Jiao-hu-dous at the front part of the Dou-gong is also large.

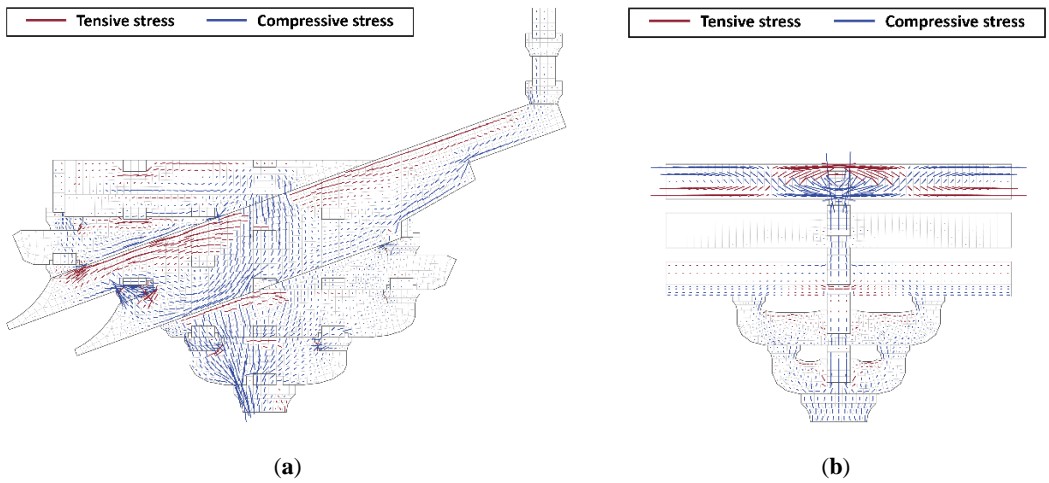

(a)　　　　　　　　　　　　　　　　(b)

**Figure 8.** Stress vector diagram of the Ji-xin-zao Dou-gong in the roof load case: (**a**) longitudinal section; (**b**) transverse section.

Figure 8b shows the load-transfer path in Fangs and Gongs, which transfers the roof load from the side to the central axis. The Yan-fang on the top of the Dou-gong mainly bears the roof load, while exhibiting a high level of stress. The Su-fang, Ni-dao-gong, and Man-gong at the bottom of the Dou-gong are subjected to the force together, and the stress is compressive. For the Tou-xin-zao Dou-gong (Figure 9a), due to the absence of the Shua-tou and Chen-fang-tou, the Angs are the only bended components in the Dou-gong when subjected to the roof load, and the regulation of the stress distribution and the load-transferring path is similar to that of the Ji-xin-zao Dou-gong. For the Fangs and Gongs in the Tou-xin-zao Dou-gong (Figure 9b), the Yan-fang and Lin-gong (at the top of the Dou-gong) bear the entire roof load together, exhibiting a high level of stress.

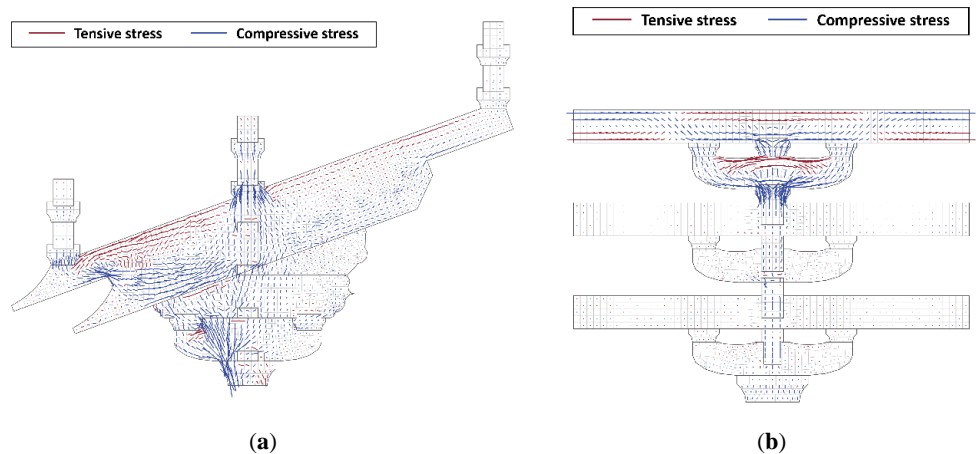

(a)　　　　　　　　　　　　　　　　(b)

**Figure 9.** Stress vector diagram of the Tou-xin-zao Dou-gong in the roof load case: (**a**) longitudinal section; (**b**) transverse section.

In the progressive collapse condition, both the Ji-xin-zao and Tou-xin-zao Dou-gong transfer the load from the central axis to the two sides through the Su-fangs (Figure 10),

whose stresses are significantly larger than that of the other components. It is remarkable that the Che-fang-tou and the middle of the Angs in the Ji-xin-zao Dou-gong are in bending. This indicates that the Chen-fang-tou can transfer the load from the Yan-fang to Liao-yan-fang, providing adequate connections among the Fangs on different Tiaos. Thus, these Fangs can share the load caused by the collapse. In the Tou-xin-zao Dou-gong, only the Su-fangs and Yan-fang at the central axis above the Lu-dou bear the load. The stress of the Liao-yan-fang on the third Tiao is small.

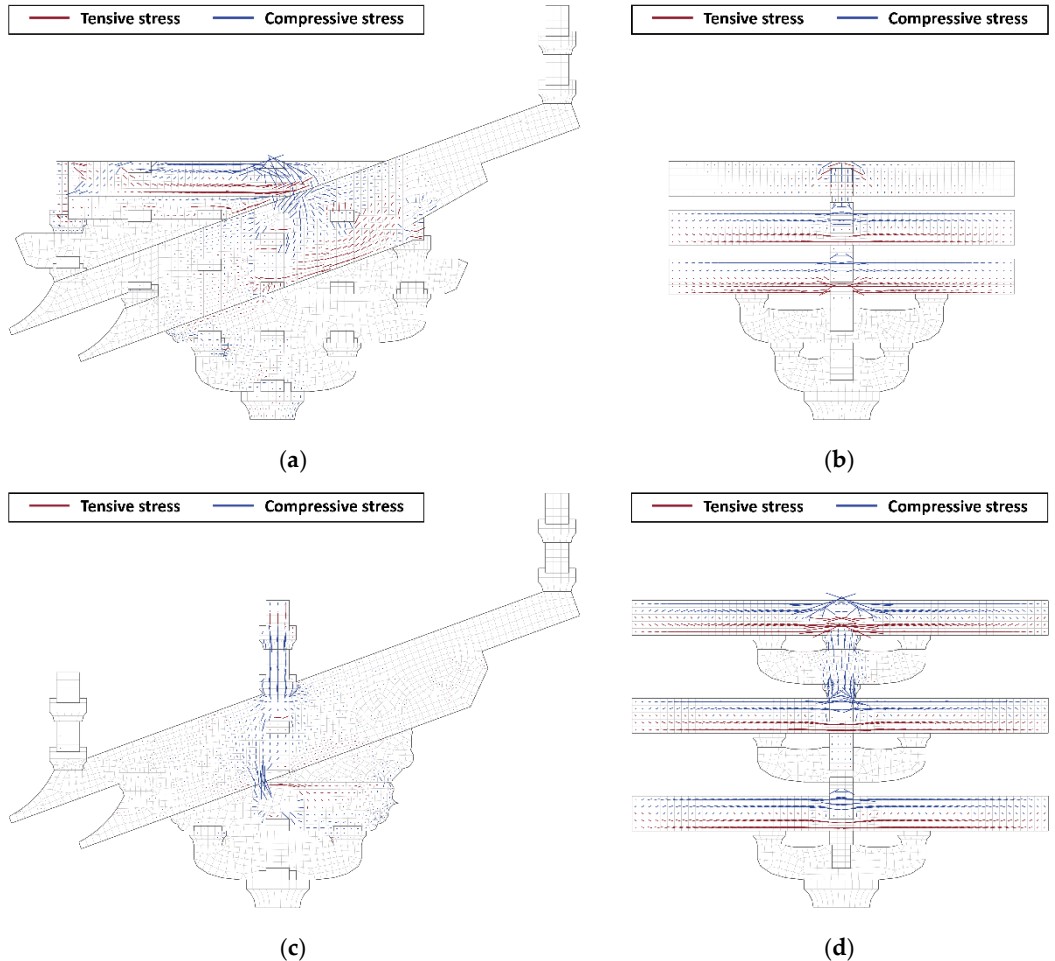

**Figure 10.** Stress vector diagram of the two Dou-gongs in the collapse case: (**a**) Ji-xin-zao, longitudinal section; (**b**) Ji-xin-zao, transverse section; (**c**) Tou-xin-zao, longitudinal section; (**d**) Tou-xin-zao, transverse section.

In summary, the load transferring paths in the two types of Dou-gongs bearing the roof load are similar. The Gongs and Fangs transfer the pressure from the two sides to the central axis. The load on the Yan-fang then continues to be transferred downward, successively passing the tail of the Chen-fang-tou and Shua-tou, the middle of the Ang, Hua-gong and Hua-tou-zi, to the Lu-dou. The load applied to the Liao-yan-fang is transferred by the front of the Angs, Hua-gong, and Hua-tou-zi. Regarding the collapse case, both the Tou-xin-zao and Ji-xin-zao Dou-gong transfer the extra load above the Lu-dou horizontally to the two sides through the Fangs, and the Gongs below have low stress.

The mechanical behavior of all the types of components in these Dou-gongs under the two conditions is listed in Table 2. For the roof load condition, the Ang is in bending, while the Hua-gong and Hua-tou-zi are generally compressed. In the Ji-xin-zao Dou-gong, the Chen-fang-tou and Shua-tou are also in bending, sharing the bending moment of the Angs. For both conditions, the Fangs are in bending. It can be noticed that the Chen-fang-tou in

Ji-xin-zao Dou-gong is also in bending when subjected to the collapse condition, providing linkages among the Fangs on different Tiaos, which makes the Fangs on different Tiaos share the load. In the Tou-xin-zao Dou-gong, due to the cancellation of the Chen-fang-tou and Shua-tou, only the Fangs above the Lu-dou bear the load in the collapse. The Gongs exhibit low stress in the collapse condition, while they usually co-work with the Fangs above in the roof load case. The Gongs with a short length, such as the Hua-gong, Gua-zi-gong and Ni-dao-gong, are mostly compressed, but the Man- and Lin-gongs, whose lengths are much larger, are in bending to some extent when the Dou-gong bears the roof load.

**Table 2.** Mechanical behavior of the components in the two Dou-gongs.

| Components | | Roof Load Case | | Collapse Case | |
|---|---|---|---|---|---|
| | | Ji-Xin-Zao | Tou-Zin-Zao | Ji-Xin-Zao | Tou-Zin-Zao |
| Gong | Hua-gong | Locally compressed | Locally compressed | Low stress | Low stress |
| | Gua-zi-gong | Compressed | – | Low stress | – |
| | Ni-dao-gong | Compressed | Compressed | Low stress | Low stress |
| | Man-gong | Bending | Bending | Low stress | Low stress |
| | Lin-gong | – | Bending | Low stress | Low stress |
| Fang | Su-fang | Bending | Bending | Bending | Bending |
| | Luo-han-fang | Bending | – | Bending | – |
| | Yan-fang | Bending | Bending | Bending | Bending |
| | Liao-yan-fang | Bending | Bending | Bending | Bending |
| Others | Xue-xie | Low stress | Low stress | Low stress | Low stress |
| | Chen-fang-tou | Bending | – | Bending | – |
| | Shua-tou | Bending | – | Low stress | – |
| | Ang | Bending | Bending | Locally bending | Locally compressed |
| | Hua-tou-zi | Locally compressed | Locally compressed | Low stress | Low stress |

## 3. A Beam-Truss Simplified Model

### 3.1. A Brief Review of Recent Simplified Approaches for Dou-Gongs

To develop a reliable simplified Dou-gong model, a review of the simplified models proposed in recent studies was firstly carried out in this section. These simplifications were briefly reviewed as follows.

Wei [38] firstly proposed a corbel model, and the Dou-gong geometric characteristics were largely simplified (the Dous were regarded as cubes, and the Gongs were regarded as cuboids). Based on that approach, Chen [39] took into account the interactions among the components of Dou-gongs in the analyses, considering the frictional energy dissipation. The accuracy of this model was experimentally verified. Yuan et al. [40] used this model to analyze typical Dou-gongs in Yin-xian Pagoda, further verifying its accuracy.

Some researchers adopted beam and spring elements for the Dou-gong simplification. Zhang [41] adopted a group of beam elements to simulate the Dou-gong mechanical behavior, calibrating the parameters through a series of on-site tests. Based on the equivalence of the dynamic characteristic, Li et al. [42] simplified Dou-gongs to beam elements with hinge or rigid joints. After carefully investigating the mechanical mechanisms of typical Dou-gongs in Yin-xian Pagoda, Chen [43] developed a "beam-spring model" as the Dou-gong simplification, and the analytical solution of the stiffness was explicitly given. Shi et al. [44] simplified the Dou-gong components to beam elements through specific regulations, verifying these simplifications through a series of experiments.

Truss elements were also employed in some studies to simulate the Dou-gong structural behavior. Liu et al. [45] claimed that the stress flows diagonally along the lines between the Dous in the Dou-gong when analyzing the structural behavior of the entrance building of Du-le Temple, and further suggested simplifying the Dou-gong entirely to a

diagonal truss. Han et al. [46] also employed truss elements to simulate the Dou-gong when investigating the wind vibration performance of the main hall of the Bao-guo Temple.

Furthermore, the spring model was most commonly employed to simulate the Dou-gong in large-scale analyses, and the stiffness parameters of the springs were usually numerically and experimentally verified (e.g., see [47,48]). However, only the entire Dou-gong performance was investigated using this model. The stress of the Dou-gong components was hard to examine.

As summarized in Table 3, spring elements can only be employed to investigate the entire Dou-gong performance, while the mechanical behavior of the Dou-gong components is hard to study in large-scale analyses. The corbel model for Dou-gongs showed high accuracy and underwent strict experimental verification and calibration, but the time consumption was high due to the adoption of solid elements. Adopting beam or truss elements certainly cuts down a large amount of time consumption. However, the geometric and construction features of Dou-gongs were not fully taken into account in the literature discussed above, and the truss sectional dimensions were not numerically verified.

**Table 3.** Typical simplified Dou-gong models in recent literature [38–48].

| Model | Element Type | Simplification | Efficiency | Consideration of the Component | Verification | Diagram |
|---|---|---|---|---|---|---|
| Corbel model | Solid | Dous are simplified as cubes, Gongs are simplified as cuboids | Low | Lu-dou, San-dou, Jiao-hu-dou, Hua-gong, Ni-dao-gong, and Man-gong | Verified through experiments |  |
| Beam element model | Beam | Fangs are simplified as beam elements, Dous are simplified as short columns | High | Su-fang and Lu-dou | Verified through natural frequency and modal |  |
| Beam-spring model | Beam, Spring | Fangs are simplified as beam elements, Dous are simplified as spring elements | High | Su-fang, Lu-dou, Jiao-hu-dou, and San-dou | Without verification |  |
| Truss model | Truss | Entirely simplified as a truss element | High | Without considering details | Without verification |  |
| Spring model | Spring | Entirely simplified as spring elements | High | Without considering details | Verified and calibrated through experiments |  |

*3.2. A New Simplified Finite Element Modeling Approach*

Based on the results obtained by the refined finite element models of the two considered Dou-gongs, a new simplified finite element Dou-gong modeling approach, which is applicable for large-scale numerical analyses, was proposed in this section. According to the literature review provided above, the computational efficiency within the application of truss/beam elements is higher than that of solid elements. Moreover, in large-scale finite element modeling of Chinese traditional timber buildings, components (e.g., columns, purlins and beams) are commonly simulated by beam elements. For these reasons, the new simplified finite element model proposed in this study consists of a series of beam and truss elements, considering "Cai-fen-zhi"—a dimension regulation in the traditional code [13]—at the same time.

The simplification was implemented by the following steps: firstly, the overall load transferring paths of tension and compression were summarized and illustrated; then, the components on the load transferring paths were simplified into beam or truss elements according to their mechanical mechanisms; finally, the connection and sectional dimension of the beam and truss elements were determined through realistic construction rules and dimension regulations.

Based on the results shown in Section 2, Figure 11 illustrates how the components of the Dou-gong transfer the load in the roof load condition. The first path of the compression lies between the Yan-fang and Liao-yan-fang, where the roof load transfers downward to the front and the middle of the Ang, respectively. After that, these two stress flows are merged at the Ang bottom, and then move diagonally along the Hua-tou-zi and Hua-gong down to the Lu-dou. Another path lies between the Yan-fang and the Lin-gong behind, through which the pressure on the Yan-fang is delivered to the Dou-gong tail, and then to the Lin-gong and purlin. Part of the stress in this path also transfers downward, going through the rear of Hua-tou-zi and Hua-gong, to the Lu-dou. There is also a third path passing through the middle of the Ang, where the pressure on the Yan-fang is transferred directly down to the Lu-dou. In addition, two compressive stress flows also pass through the bottom of the Chen-fang-tou and Shua-tou, respectively. Thus, with the consideration of the Dou-gong construction rules and dimensional regulations, the new beam-truss simplified model was proposed as follows (Figure 11b): the Chen-fang-tou and Shua-tou were simplified as beam elements, with sectional dimensions $b \times h_1$; two Angs were regarded as one beam element, with a section of $b \times (h_1 + h_2)$; the Hua-gong and Hua-tou-zi were simulated by truss elements because of the compression dominance. Because of the difference in the bending stiffness of the beam element and the axial stiffness of the truss element, it was necessary to modify the sectional dimensions when the truss element was used to simulate the bending behavior. Thus, the sectional area of the truss element is $k_1 \times b \times h_2$, where $k_1$ is the reduction factor to consider this equivalence; the Shua-tou at the Dou-gong rear was simplified to a beam ($b \times h_1$), connecting with the Shua-tou at the front; square beams ($b \times b$) were used to simulate vertical linkages.

Figure 11c,d illustrates how Gongs and Fangs transfer the roof load. There are mainly two Gong arrangements, which are as follows: the Dan-gong (one Ni-dao-gong/Gua-zi-gong and one Su-fang) and Chong-gong (one Ni-dao-gong, one Man-gong, and one Su-fang). For the Dan-gong, the pressure on the Su-fang moves downward. The load on the two sides of the Fang passes through the bottom of the Gong and then moves toward the central axis; the load in the middle is transferred downward directly. For the Chong-gong, part of the pressure on the Su-fang is transferred diagonally downward to the Lu-dou through the two sides of the Man-gong and Ni-dao-gong. Another part of the stress flows along the central axis and downward to the Lu-dou. Based on these analyses, the new simplified beam-truss model for these two kinds of combinations is illustrated in Figure 11e,f. For the Dan-gong, the Ni-dao-gong/Gua-zi-gong was simulated by two truss elements with a reduced sectional area ($k_2 \times b \times h_1$), where $k_2$ is the reduction factor. For the Chong-gong, the Man-gong and Ni-dao-gong/Gua-zi-gong were entirely simulated by two truss elements, with a sectional area of $k_3 \times b \times h_1$, where $k_3$ is the corresponding

reduction factor. The load transferring paths at the central axis in the two cases above were regarded as square beams ($b \times b$).

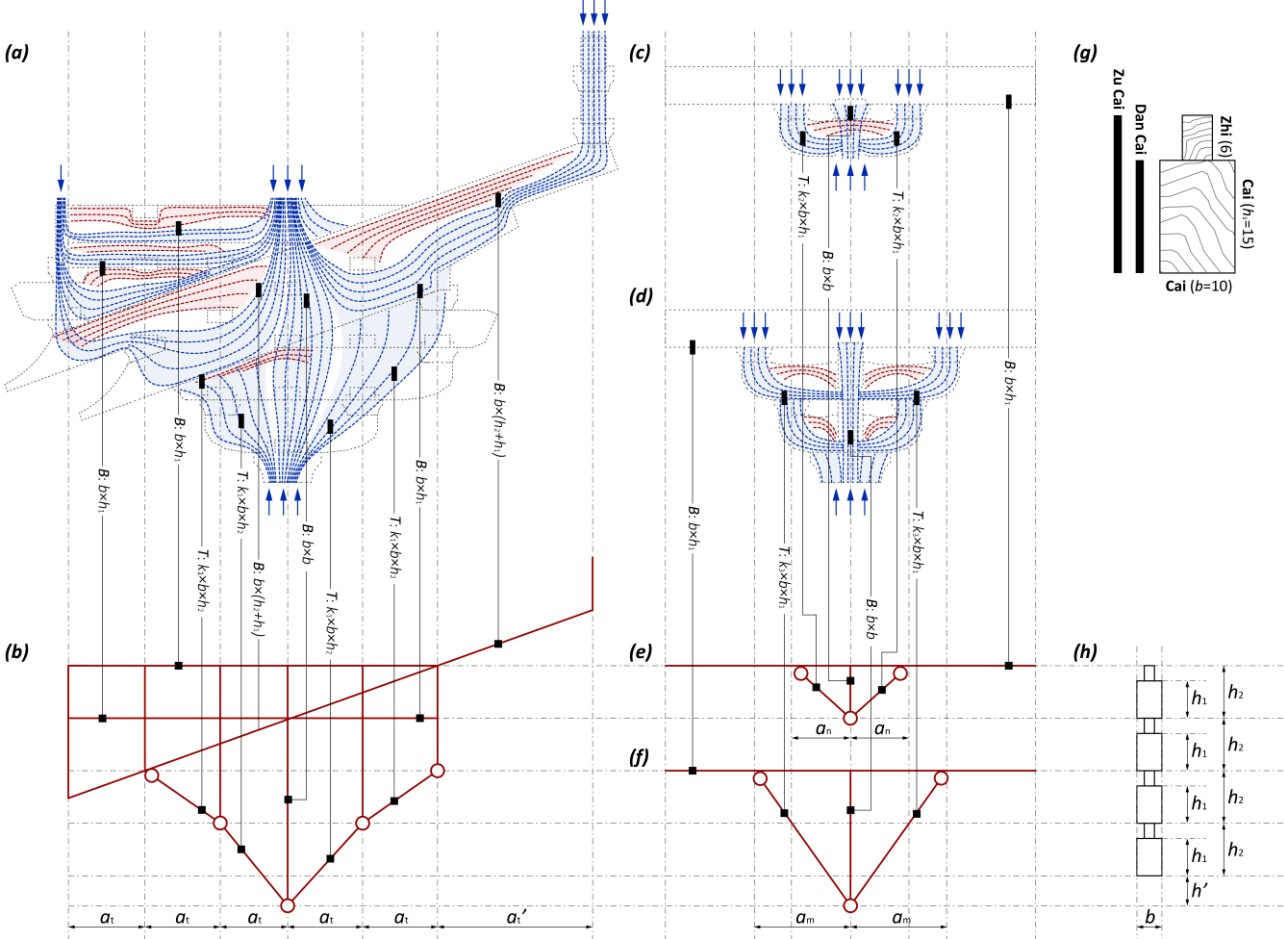

**Figure 11.** Overview of the Dou-gong simplified modeling approach. **Note:** the vertical line connects the component before and after the simplification. The labels at the middle of these lines represent the section information: ***B*** represents the beam element and ***T*** represents the truss element; the expression afterward represents the sectional dimension. The meaning of other symbols is listed as follows: $b$ is the Cai width; $h_1$ is the Dan Cai height; $h_2$ is the Zu Cai height; $a_t$ is the Tiao length; $a_m$ is the Man-gong half-length; $a_n$ is the Ni-dao-gong half-length; $k_1$ is the reduction factor of the Hua-gong sectional area; $k_2$ is the reduction factor of the Dan-gong sectional area; $k_3$ is the reduction factor of the Chong-gong sectional area.

Under the collapse condition, as shown in Table 3, the mechanical behavior of the components in the Dou-gong is quite similar to that in the roof load condition. The only difference is that the Fangs are separated from the Gongs and bear the load individually in the collapse condition, resulting in low stress of the Gongs. To take into account this situation, both the trusses and vertical square beams ($b \times b$) mentioned above were considered as compressed only.

The new simplified model proposed above involves several reduction factors ($k_1$, $k_2$, and $k_3$) for the truss sectional area, and their analytical expressions were established based on the principle of equal-rigidity substitution, as shown in the below formulation derivation.

To obtain the reduction factors $k_1$ and $k_2$, the structural behavior of a single Gong was firstly investigated (Figure 12a). The half of the Gong can be regarded as a cantilever beam (length $a$), with a concentrated load at the free end (Figure 12b). Based on the elastic theory,

the vertical stiffness of the cantilever beam and diagonal truss, $K$ and $K'$, respectively, is as follows:

$$K = \frac{3EI}{a^3} \tag{1}$$

$$K' = \frac{EA\sin^2\theta\cos\theta}{a} \tag{2}$$

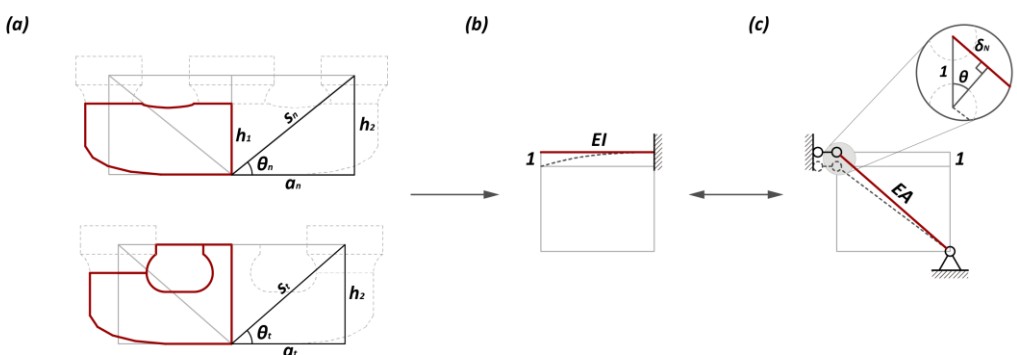

**Figure 12.** Calculation flow of the sectional reduction factors for the Dan- and Hua-gongs: (**a**) dimension of the Dan- and Hua-gong; (**b**) vertical stiffness of a cantilever beam; (**c**) vertical stiffness of the equivalent truss element.

Based on the rigid-equal principle $K = K'$, the truss area $A$ can be determined by the following equation:

$$A = \frac{3I}{a^2\sin^2\theta\cos\theta} \tag{3}$$

According to the Yin-zao-fa-shi [13], a regulation for the construction in ancient China, for the Hua-gong and Gua-zi-gong/Ni-dao-gong, the Zu Cai and Dan Cai were adopted, respectively. For these two cases, the following substitution of geometric parameters with different sectional height was implemented, with $h_1$ (height of the Dan Cai) for the Ni-dao-gong/Gua-zi-gong and $h_2$ (height of the Zu Cai) for the Hua-gong:

$$I = \frac{bh_2^3}{12}, \; A = k_1bh_2, \; \theta = \theta_t, \; \cos\theta = \frac{a_t}{s_t}, \; \sin\theta = \frac{h_2}{s_t}, \; a = a_t \tag{4}$$

$$I = \frac{bh_1^3}{12}, \; A = k_2bh_1, \; \theta = \theta_t, \; \cos\theta = \frac{a_n}{s_n}, \; \sin\theta = \frac{h_2}{s_n}, \; a = a_n \tag{5}$$

The reduction factors $k_1$ and $k_2$ can be obtained as follows:

$$k_1 = \frac{s_t^3}{4a_t^3} = \frac{1}{4\cos\theta_t^3} \tag{6}$$

$$k_2 = \frac{h_1^2 s_n^3}{4h_2^2 a_n^3} = \left(\frac{h_1}{h_2}\right)^2 \frac{1}{4\cos\theta_n^3} \tag{7}$$

Specifically, based on the Yin-zao-fa-shi [13], the Zu Cai height is 21 Fens; the Dan Cai height is 15 Fens; the width of the Zu Cai and Dan Cai are both 10 Fens; the Tiao of the Hua-gong should not exceed 30 Fens; the length of the Ni-dao-gong/Gua-zi-gong is 62 Fens. Thus, substituting these values in Equations (6) and (7), the values of the reduction factors $k_1$ and $k_2$ are equal to 0.441 and 0.227, respectively.

As for the Chong-gong, which is a combination of the Man- and Gua-zi-gongs (Figure 13a), the half of the Chong-gong can be regarded as two cantilever beams connected by a vertical rigid rod, where at the free end of the upper beam a concentrated load was applied (Figure 13b). For both beams, the Dan Cai was adopted, with a section of $b \times$

$h_1$. To solve the stiffness of the whole structure, a unit displacement $\delta_A = 1$ was applied at the free beam end.

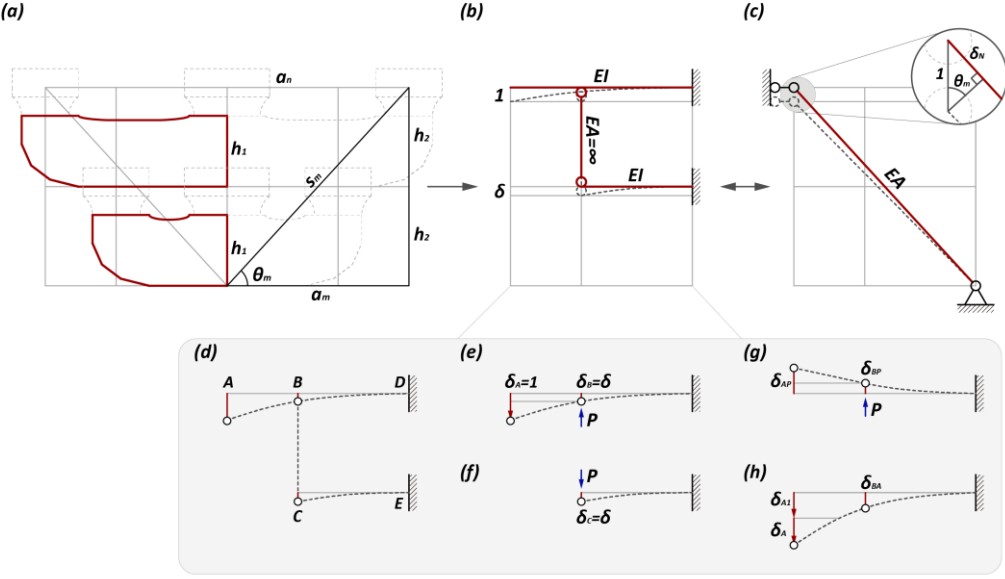

**Figure 13.** Calculation flow of the sectional reduction factor for the Chong-gong: (**a**) dimension of the Chong-gong; (**b**) diagram of the beam model of the Chong-gong; (**c**) vertical stiffness of the equivalent truss element; (**d**) deformation of the beam model; (**e**) force diagram for the upper beam; (f) force diagram of bottom beam; (**g**) deformation of the upper beam subject to force P; (**h**) deformation of the upper beam subject to concentrate load at the end.

The deformation diagram of the structure, in this case, is shown in Figure 13d. Replacing the rigid rod with a pair of balanced forces $P$, the structure can be simplified into two independent beams (Figure 13e,f). For the beam in Figure 13f, the relationship between the vertical displacement $\delta_C$ at the free end and the axial force of the rigid rod P can be written as the following equation:

$$P = \frac{3EI}{a_n^3}\delta_C \tag{8}$$

The case illustrated in Figure 13e can be regarded as a superposition of the cases shown in Figure 13g,h. The relationship between the vertical displacement $\delta_B$ at point B and the external load can be solved through the superposition principle and graphic multiplication, which is demonstrated by the following equation:

$$\delta_B = \delta_{BA} - \delta_{BP} = \frac{a_n^2(3a_m - a_n)}{2a_m^3}\left(1 + \frac{Pa_n^3}{3EI} + \frac{Pa_n^2(a_m - a_n)}{2EI}\right) - \frac{Pa_n^3}{3EI} \tag{9}$$

According to the rigid-rod assumption, the deformations at points B and C are the same, i.e., $\delta_B = \delta_C = \delta$. Thus, the deformation $\delta$ can be obtained by the following equation:

$$\delta = \frac{2a_n^2(3a_m - a_n)}{8a_m^3 - a_n(3a_m - a_n)^2} \tag{10}$$

The vertical stiffness of the whole structure, $K$, is the sum of the vertical reactions of two supports (points D and E), which can be solved as follows:

$$
\begin{aligned}
K &= R_{VD} + R_{VE} \\
&= \frac{3EI}{a_m^3}\left(1 + \frac{Pa_n^3}{3EI} + \frac{Pa_n^2(a_m-a_n)}{2EI}\right) \\
&= \frac{3EI}{a_m^3}\left(1 + \delta\frac{3a_m-a_n}{2a_n}\right) \\
&= \frac{24EI}{8a_m^3 - a_n(3a_m-a_n)^2}
\end{aligned}
\tag{11}
$$

The vertical stiffness of the equivalent diagonal truss can be written as follows (Figure 13c):

$$
K' = \frac{EA sin^2\theta_m cos\theta_m}{a_m}
\tag{12}
$$

According to the rigid-equal principle $K = K'$, the area of the equivalent truss can be solved using the following equation:

$$
A = \frac{24Ia_m}{\left(8a_m^3 - a_n(3a_m - a_n)^2\right)sin^2\theta_m cos\theta_m}
\tag{13}
$$

By substituting the geometric relations in Equation (14), the analytical solution of the reduction factor $k_3$ can be written as Equation (15).

$$
I = \frac{bh_1^3}{12}, \ A = k_3 bh_1, \ cos\theta_m = \frac{a_m}{s_m}, \ sin\theta_m = \frac{2h_2}{s_m}
\tag{14}
$$

$$
k_3 = \frac{h_1^2 s_m^3}{2h_2^2\left(8a_m^3 - a_n(3a_m - a_n)^2\right)}
\tag{15}
$$

Similarly, based on the Yin-zao-fa-shi [13], the Dan Cai height is 15 Fens; the Dan Cai width is 10 Fens; the length of the Ni-dao-gong/Gua-zi-gong is 62 Fens; the Man-gong length is 92 Fens. Thus, by substituting these values in Equation (15), the value of the reduction factor $k_3$ equal to 0.141 is obtained.

By observing Equations (6), (7) and (15), it was found that the value of the reduction factor is related to the angle of the truss and the Gong sectional dimension; both a larger slope angle (e.g., the Chong-gong) and a larger sectional dimension (e.g., the Hua-gong) of the equivalent truss lead to smaller reduction factors. Moreover, factors $k_1$, $k_2$, and $k_3$ are smaller than one, indicating that the sectional area must be reduced when introducing equivalent truss elements. Otherwise, the stiffness of the model will be incorrectly increased.

### 3.3. Numerical Verification

According to the Dou-gong beam-truss simplification proposed, the refined models of the two Dou-gongs from Section 2 were simplified into beam-truss finite element models. The results of the simplified models were compared with those of the refined ones, including their stiffness, deformations, and stress, in order to perform a verification. Moreover, the time consumption of the simplified and refined models was compared.

The simplified models of the two Dou-gongs are shown in Figure 14. The load condition and material parameters of these two models are consistent with those of the refined models from Section 2. Both the roof load and collapse conditions investigated in Section 2 were taken into account. For the roof load condition, besides the supports imposed in the refined model, several rotational free degrees of the beam elements are fixed; only XZ- and XY-in-plane rotation are allowed for the two ends of the Fangs and the Ling-gong on the tail of Ang, respectively (Figure 14a,b). For the collapse condition, the

ends of the Fangs and the Ling-gong on the tail of Ang are vertically fixed (Figure 14c,d). The simplified models were analyzed through the ABAQUS/Standard module.

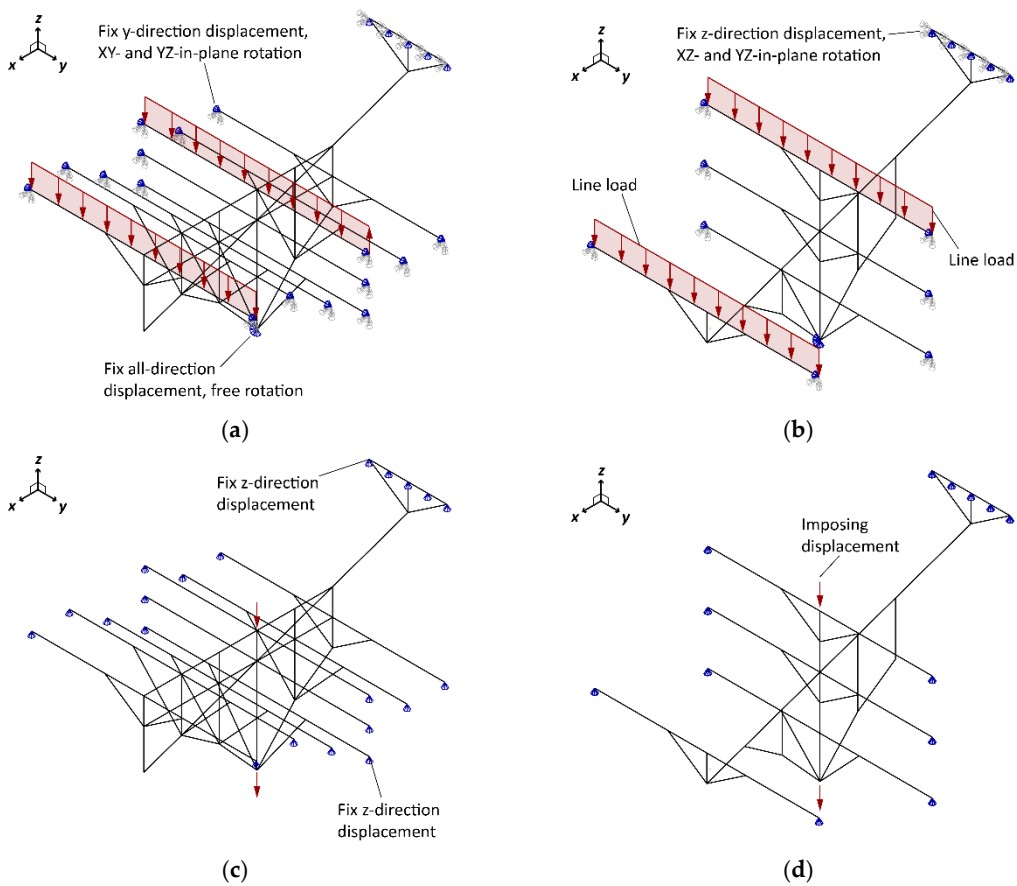

**Figure 14.** Boundary conditions of the simplified models for the two Dou-gongs: (**a**) Ji-xin-zao, roof load condition; (**b**) Tou-xin-zao, roof load condition; (**c**) Ji-xin-zao, collapse condition; (**d**) Tou-xin-zao, collapse condition.

Firstly, the consistency of stress distribution was investigated. For the roof load condition, as illustrated in Figure 15, the stress distribution in the refined and simplified models is similar. In the Ji-xin-zao Dou-gong, the stress of the Yan-fang and the stress near the central axis are higher than that of other components. In the Tou-xin-zao Dou-gong, the Yan-fang, Lin-gong (at the top), and the front of the Angs exhibit an obvious high level of stress. Moreover, the value of the Mises stress in the refined model and the corresponding simplified model only has a small difference (1.3–3.7%). These consistencies indicate that the simplification for the Dou-gong proposed in this study is accurate for the roof load condition.

In the collapse condition, the refined model and corresponding simplified model of the two Dou-gongs also lead to a similar stress distribution (Figure 16), where the Fangs are the main load-bearing components. In these models, the Fangs at the central axis of the Dou-gong exhibit larger stress, and the maximum stress is found in the middle of the Yan-fang. The stress of the Chen-fang-tou in the refined and simplified models for the Ji-xin-zao Dou-gong is also large. Regarding the value of the stress in the refined and simplified models, in the Tou-xin-zao Dou-gong, the maximum value is quite close (about 1.8%), while in the Ji-xin-zao Dou-gong, slight differences in the maximum value appear (about 15.6%). This is due to the stress concentration in the refined model, caused by the non-smooth geometric characteristics of the mortise-tenon joint.

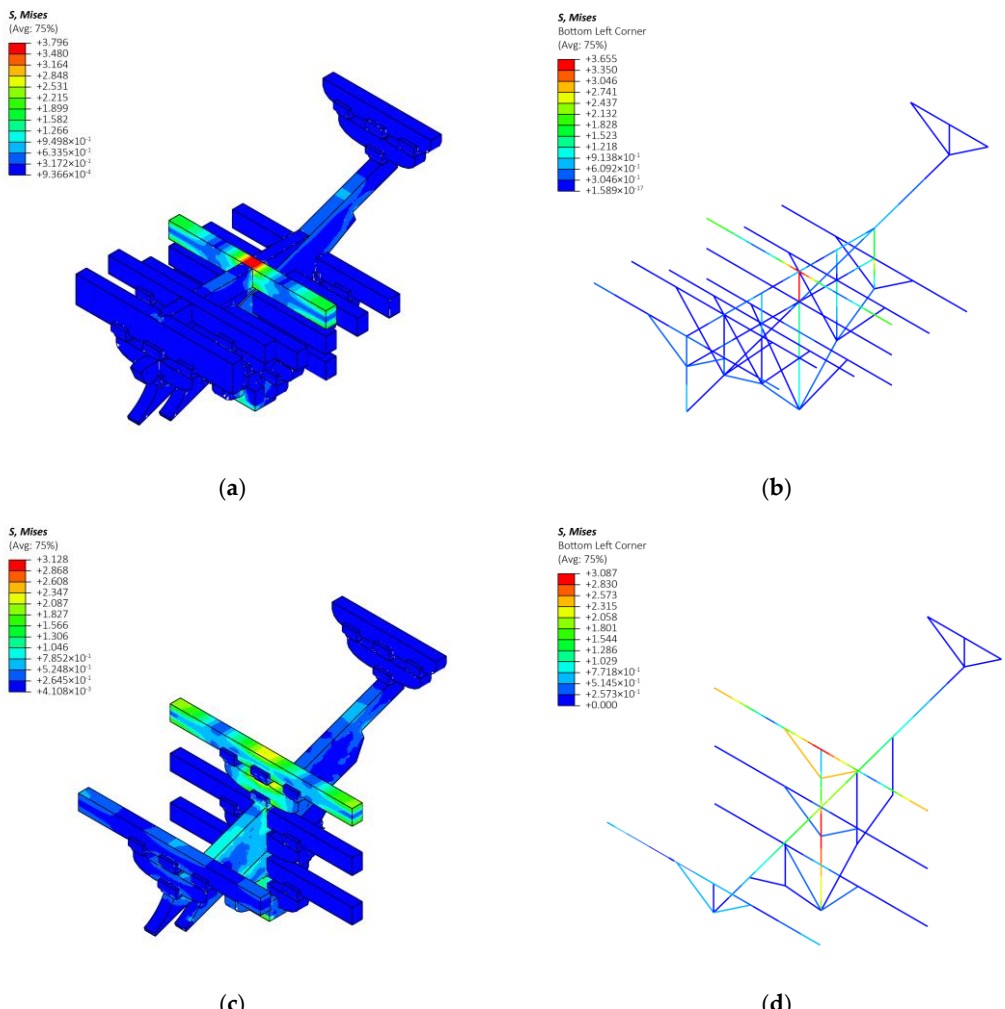

**Figure 15.** Mises stress of the Dou-gongs bearing the roof load, refined model vs. simplified model (unit: MPa): (**a**) refined model, Ji-xin-zao Dou-gong; (**b**) simplified model, Ji-xin-zao Dou-gong; (**c**) refined model, Tou-xin-zao Dou-gong; (**d**) simplified model, Tou-xin-zao Dou-gong.

Secondly, the deformation results are compared (Figure 17). For the roof load condition, both the refined and simplified models of the Ji-xin-zao Dou-gong entirely lean forward and the position of the corresponding components after the deformation is consistent with each other. For the Tou-xin-zao Dou-gong bearing the roof load, although the overall tendency of the deformation of the two models is the same, the position of the Angs in the two models presents some differences. Namely, an obvious slide between the Ang and the Hua-tou-zi appears in the refined model, while the relative displacement among the components in the simplified model is small (Figure 17c). This difference is because the connections between the beam and truss elements are hinge joints, where large relative displacements are not allowed. Nevertheless, the difference in the maximum vertical displacement of the refined and simplified models is quite limited and is 4.6–10.7%. The deformation of the refined and simplified models in the collapse condition is similar. The Chen-fang-tous and Shua-tous in the two Ji-xin-zao Dou-gongs are bending upward; two Tou-xin-zao Dou-gongs move entirely downwards and the positions of the components after the deformation in these two models are generally consistent with each other. It is also worthwhile mentioning that, both in the Ji-xin-zao and Tou-xin-zao Dou-gong, the separations among the components (e.g., the tail of the Ang and Lin-gong above) are successfully simulated through compressed-only trusses.

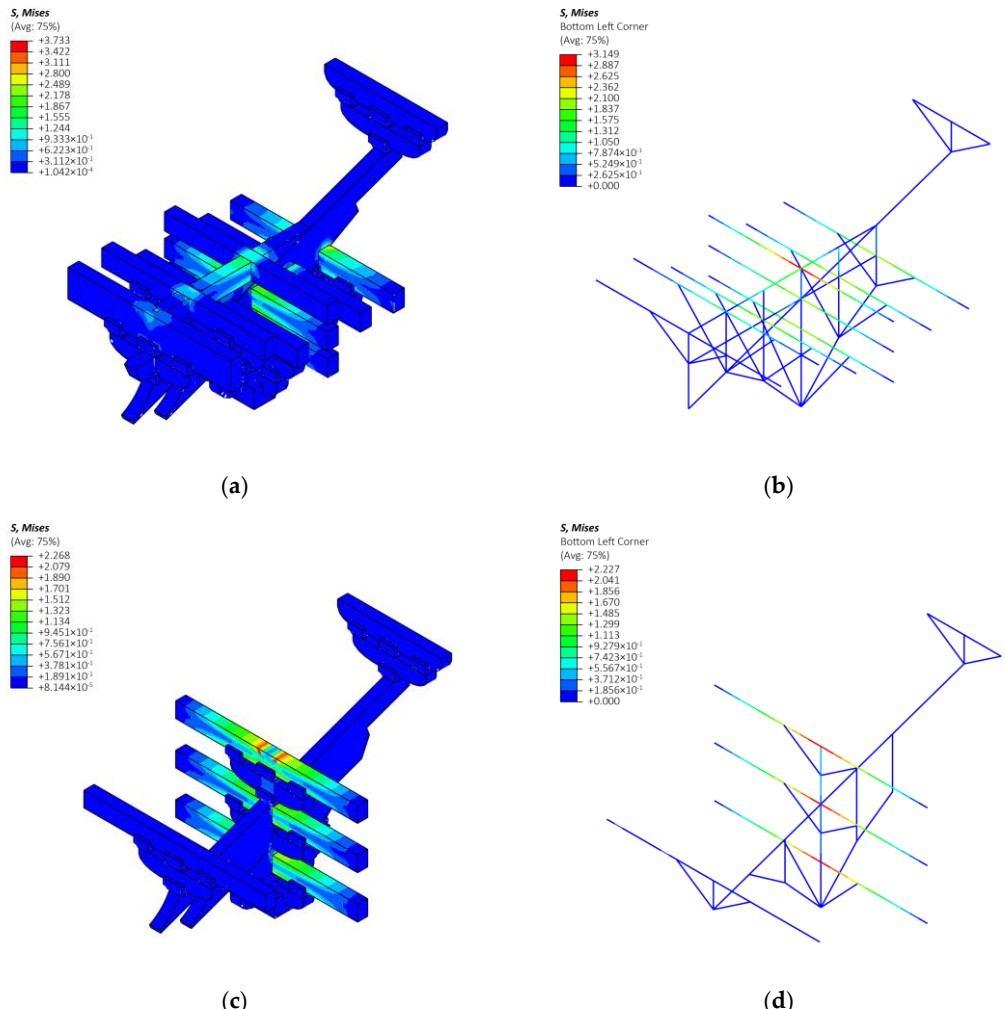

**Figure 16.** Mises stress of the Dou-gongs in the collapse case, refined model vs. simplified model (unit: MPa): (**a**) refined model, Ji-xin-zao Dou-gong; (**b**) simplified model, Ji-xin-zao Dou-gong; (**c**) refined model, Tou-xin-zao Dou-gong; (**d**) simplified model, Tou-xin-zao Dou-gong.

Then, the vertical stiffness of the refined and simplified models is also investigated. As shown in the load–displacement curve (Figure 18a), the vertical stiffness of the Ji-xin-zao Dou-gong is quite larger than that of the Tou-xin-zao Dou-gong when bearing the roof load. For the Ji-xin-zao Dou-gong, the vertical stiffness of the refined model and the simplified model coincided with the linear load increase. For the Tou-xin-zao Dou-gong, the load–displacement curves of the refined and simplified models show some differences. The load of the refined model increases non-linearly, while the load–displacement curve of the simplified model is almost linear. This difference is caused by the lack of connections among the components in the Tou-xin-zao Dou-gong. The relative slide happens in the refined model, leading to a non-linear displacement increase during the loading. Although the initial tangential stiffness of these two models is different, the secant modulus near the peak is similar, indicating that the simplified model is still applicable for this condition. For the collapse condition (Figure 18b), the vertical stiffness of the Ji-xin-zao Dou-gong is still larger than that of the Tou-xin-zao Dou-gong. In this condition, all load–displacement curves are linear. For both the Ji-xin-zao and Tou-xin-zao Dou-gong, the load–displacement curves of the refined and simplified models show great consistency.

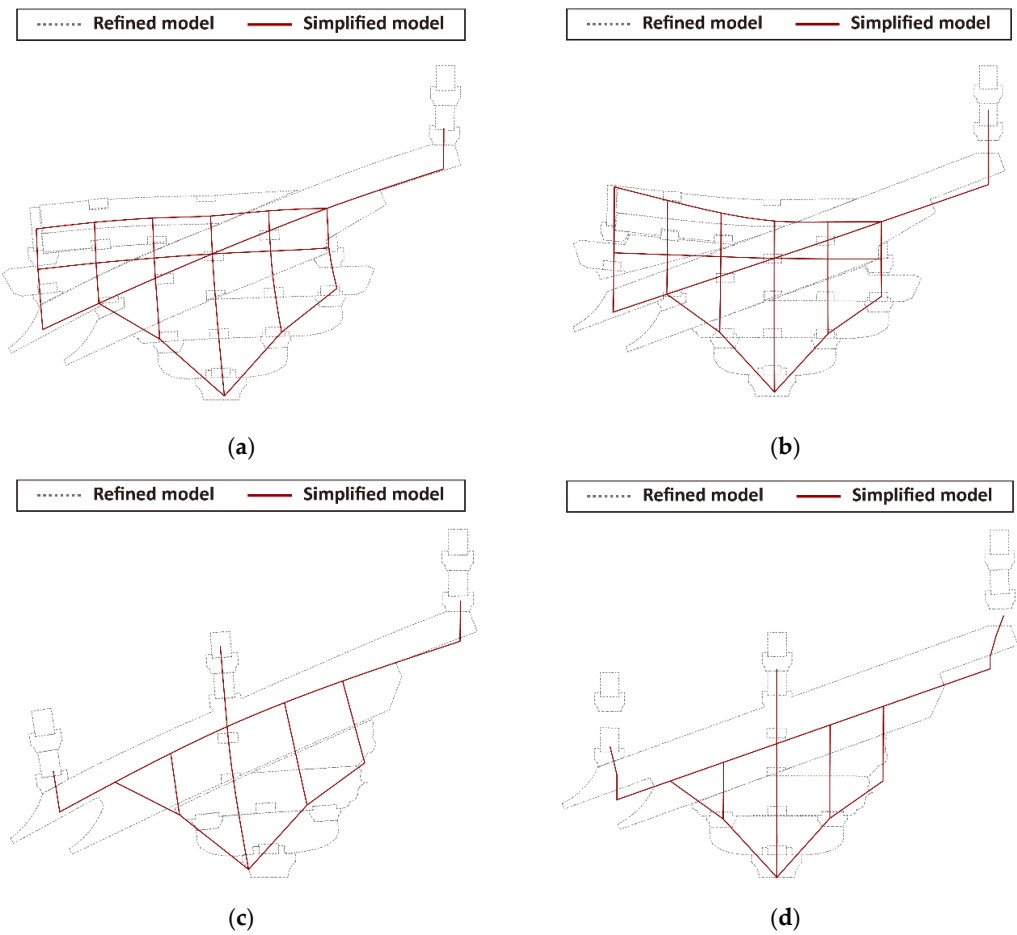

**Figure 17.** Comparison of the deformation results, refined model vs. simplified model: (**a**) Ji-xin-zao Dou-gong in the roof load condition; (**b**) Ji-xin-zao Dou-gong in the collapse condition; (**c**) Tou-xin-zao Dou-gong in the roof load condition; (**d**) Tou-xin-zao Dou-gong in the collapse condition.

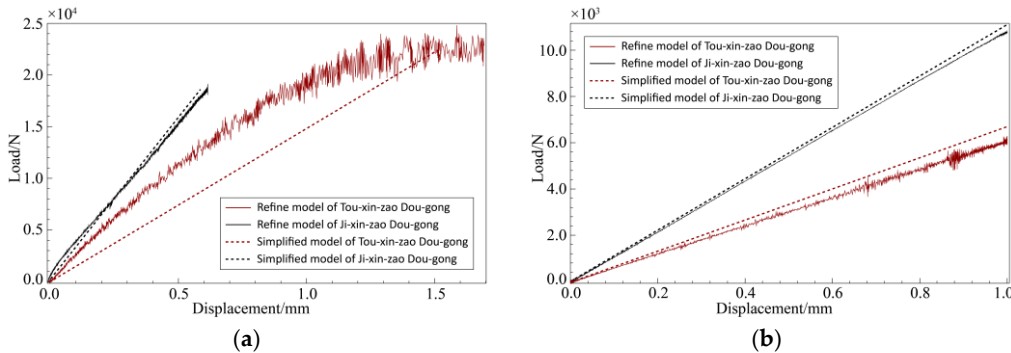

**Figure 18.** Comparison of the load–displacement curves, refined model vs. simplified model: (**a**) roof load condition; (**b**) collapse condition.

Finally, the time consumptions for the computation of the refined and simplified models are compared and listed in Table 4. The results show that the computation of the simplified model requires noticeably less time, with a drop of 90–97%; the time consumption of the refined model is about 6–12 min, while for the simplified model it is 13–18 s. This proves that adopting beam and truss elements can significantly improve the efficiency of numerical analyses.

**Table 4.** Comparison of the time consumptions: refined model vs. simplified model.

| Condition | Jin-Xin-Zao Dou-Gong | | | Tou-Xin-Zao Dou-Gong | | |
|---|---|---|---|---|---|---|
| | Refined Model/s | Simplified Model/s | Diff./% | Refined Model/s | Simplified Model/s | Diff./% |
| Roof load | 488 | 13 | 97.3 | 724 | 18 | 97.5 |
| Progressive collapse | 459 | 15 | 90.2 | 367 | 17 | 95.4 |

In summary, the numerical test results of the Ji-xin-zao Dou-gong and Tou-xin-zao Dou-gong before and after the simplification, including stress, deformation, stiffness, and time consumption, are listed in Table 5. The stress distribution of the two models is generally in agreement, and the difference of the maximum stress is 1.3–15.6%. The deformation of the simplified model of the Dou-gongs is also consistent with that of the refined model and the difference of the maximum displacement is 4–10%. For the vertical stiffness, the accuracy of the simplification of the Ji-xin-zao Dou-gong is better; the difference of the tangential stiffness and secant stiffness is about 11% and 4%, respectively. Due to the slide among the components in the Tou-xin-zao Dou-gong, the model before and after the simplification exhibits unneglectable differences in the tangential stiffness, while the difference in the secant stiffness is only around 9%. The time consumption of the model after the simplification significantly decreased, with a drop of 97%. As a result, the simplified finite element modeling approach proposed in this study shows a high computational efficiency with acceptable numerical accuracy in most cases, adapted to large-scale finite element analyses for Chinese traditional timber buildings.

**Table 5.** Summary of the results of the two Dou-gongs: refined model vs. simplified model.

| | | | Max. Stress/ MPa | Max. Vertical Displace-ment/mm | Vertical Stiffness | | Time Consumption/s |
|---|---|---|---|---|---|---|---|
| | | | | | Tangential [1] /$10^4$N mm$^{-1}$ | Secant [2] /$10^4$N mm$^{-1}$ | |
| Ji-xin-zao Dou-gong | Roof load condition | Refined | 3.796 | −0.614 | 3.566 | 3.054 | 488 |
| | | Simplified | 3.655 | −0.586 | 3.175 | 3.175 | 13 |
| | | Diff. | 3.7% | 4.6% | 11.0% | 4.0% | 97.3% |
| | Collapse condition | Refined | 3.733 | – | 1.007 | 1.079 | 459 |
| | | Simplified | 3.149 | – | 1.111 | 1.111 | 15 |
| | | Diff. | 15.6% | – | 10.3% | 3.0% | 90.2% |
| Tou-xin-zao Dou-gong | Roof load condition | Refined | 3.128 | −1.693 | 2.722 | 1.513 | 724 |
| | | Simplified | 3.087 | −1.512 | 1.482 | 1.482 | 18 |
| | | Diff. | 1.3% | 10.7% | 45.6% | 2.0% | 97.5% |
| | Collapse condition | Refined | 2.268 | – | 0.579 | 0.613 | 367 |
| | | Simplified | 2.227 | – | 0.668 | 0.668 | 17 |
| | | Diff. | 1.8% | – | 15.4% | 9.0% | 95.4% |

**Note:** [1] "Tangential" represents the tangential stiffness of the structure at the origin, i.e., the tangent slope of the load–displacement curve at the origin; [2] "secant" represents the secant stiffness of the structure at the peak of the load–displacement curve.

## 4. Conclusions

The simplified Dou-gong finite element modeling approach of early Chinese traditional timber buildings was studied. Firstly, the features of these early Dou-gongs were summarized through a literature review. Then, the refined finite element analyses of typical Dou-gongs were conducted, through which the load transferring path and the mechanical performances of Dou-gongs were investigated and summarized. Based on the obtained results, a new beam-truss model for the Dou-gong was proposed, and the analytical solution for the sectional dimensions of the equivalent truss in this simplification was established, according to the equal-rigid principle. Finally, the numerical verification of the simplified model was implemented, comparing the deformation, stress, stiffness,

and time consumption of the finite element model before and after the simplification. The conclusions were drawn as follows:

- For Dou-gongs bearing the roof load, there are two load transferring paths, which are as follows: the load applied to the Yan-fang at the central axis is transferred downwards successively through the middle of the Ang, Hua-tou-zi, and Hua-gong; the pressure on the Liao-yan-fang is diagonally delivered to the Lu-dou, passing through the front of the Ang, Hua-tou-zi, and Hua-gong. The Chen-fang-tou and Shua-tou in the Ji-xin-zao Dou-gong bend together with the Angs;

- In the collapse condition, the load in the middle of the Dou-gong is transferred to the two sides through the Fangs. The Chen-fang-tou and Shua-tou in the Ji-xin-zao Dou-gong provided efficient connections among the different Fangs, which makes them share the load. However, in the Tou-xin-zao Dou-gong, only the Fangs at the central axis above the Lu-dou bear the load;

- A new beam-truss simplified model for the Dou-gong is proposed herein based on the results of the load transferring path obtained from the refined finite element analyses. When simplifying components into truss elements, the sections of truss elements have to be reduced, compared to the original sections. The analytical solution of the reduction factors is proposed based on the equal-rigidity principle;

- Compared with the results of the refined model, the simplified model proposed in this study shows an acceptable computational accuracy concerning stress, deformation, and stiffness, with a 90–97% reduction in the time consumption, which is suitable for applications in large-scale structural simulations of early Chinese traditional timber buildings.

**Author Contributions:** Y.H. contributed to the conceptualization, numerical simulation, on-site investigation, result analysis, and original draft of this manuscript; Q.C. contributed to the conceptualization, methodology, and review of this manuscript; X.J. contributed to the on-site investigation. All authors have read and agreed to the published version of the manuscript.

**Funding:** This research was funded by the National Natural Science Foundation of China, Grant No. 51778122.

**Institutional Review Board Statement:** Not applicable.

**Informed Consent Statement:** Not applicable.

**Data Availability Statement:** Not applicable.

**Conflicts of Interest:** The authors declare no conflict of interest.

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
