# Peer review of "Simplified Calculation Model for Typical Dou-Gong Exposed to Vertical Loads"

_buildings, doi:10.3390/buildings12050689_

Round 1

Reviewer 1 Report

An interesting analysis of calculation modelling Dou-gong as specific structural segment of ancient, traditional Chinese timber buildings is presented in the paper. Based on the review of different Duo-gong types features and examination of mechanical behaviour of two selected types of these complex structural segments, under two specific vertical load cases, by refined FEM analysis (with solid elements), a new, simplified calculation modelling approach was elaborated. Models for two types of Dou-gong were developed and tested. The accuracy of proposed beam-truss models was numerically verified, confirming their reliability and efficiency from the aspect of time consumption. The same modelling approach may be used for other Dou-gong types in traditional architecture of Far East.

It would be interesting (for the continuation of this study – in the next research phase) to analyse behaviour of other wood species, i.e. various mechanical characteristics of Dou-gong material.

This is an interesting and well written paper, exhibiting an excellent command of English, as well as very systematic research and writing style. The paper topic and content are appropriate for its publication in journal Buildings, especially in its special issue Advances in Building Conservation. The manuscript is very well prepared and technically edited. The writing is clear and well-organized. The obtained results and conclusions are commented in a clear and in an exhaustive manner, with mainly excellent graphical representation. Wide range of literature was referenced.

I recommend accepting the paper, after minor revision, mostly of technical character.

Comments:

1/ Recommendation regarding the writing “Dou-gong”/”Dougong”: Although both ways are present in literature, it would be nice to keep only one type throughout the whole paper – either Dou-gong or Dougong (i.e. with or without hyphen).

2/ Please, check the sentence in lines 62-64, from the language aspect (repeated group of words).

3/ Although the Figure 2 is followed by appropriate textual explanation, it might be of help to include more details from text into the Figure 2 itself, so to be more obvious.

4/ Some photos are too dark and may not adequately represent what is intended to be pictured – readers may not clearly see what authors seen on site and wanted to show. It would be good if they may be replaced by lighten and clearer photos.

5/ The legends of diagrams in Figure 18 should be revised. According to the text describing these diagrams and drawing certain conclusions from them, it seems that terms “Ji-xin-zao” (should be black?) and “Tou-xin-zao” (should be red?) has been switched in legends. Furthermore, for both lines representing simplified model (black and red) is written “Tou-xin-zao”, while one (black?) should be “Ji-xin-zao”.

6/ In Table 5, a symbol “%” is missing in values of difference in column “Time consumption”.

7/ Splitting figures and tables on two pages (figure/table starts at the end of one page and continues at the beginning of the next page) should be avoided if possible – in case of small figures/tables that may easily fit into one page. In particular the situation to have only table caption on one page and the whole table on the next page is not convenient (e.g. Table 3).

8/ Section “Author Contributions”, at the end of paper: Please, refine authors’ names/initials, so to be clearly recognisable among the authors’ names listed in the first page of the paper.

Reviewer 2 Report

The paper is interesting. The topic worth to be study and fits the journal aims and scope. The research is well (methodology) conducted and the conclusions are based on reliable results. Even if we can see past researches on this specific timber joints, very complex, it was the first time that I saw an attempt to simplified if and to propose some guidelines useful for practitioners. 

In attachment you can find more detailed comments. My strong comment: figures and tables must be introduced by the text. Some figues simply appear and, the reader, doesn't undertand why the figure is there.... please correct it throughout the document.
